# Water vapour adjustments and responses differ between climate drivers

Øivind Hodnebrog[1], Gunnar Myhre[1], Bjørn H. Samset[1], Kari Alterskjær[1], Timothy Andrews[2], Oliviér Boucher[3,4], Gregory Faluvegi[5,6], Dagmar Fläschner[7], Piers M. Forster[8], Matthew Kasoar[9,10], Alf Kirkevåg[11], Jean-Francois Lamarque[12], Dirk Olivié[11], Thomas B. Richardson[8], Dilshad Shawki[9], Drew Shindell[13], Keith P. Shine[14], Philip Stier[15], Toshihiko Takemura[16], Apostolos Voulgarakis[9], and Duncan Watson-Parris[15]

[1]CICERO Center for International Climate Research, Oslo, Norway.
[2]Met Office Hadley Centre, Exeter, UK.
[3]Institut Pierre-Simon Laplace, Paris, France.
[4]CNRS / Sorbonne Université, Paris, France.
[5]NASA Goddard Institute for Space Studies, New York, USA.
[6]Center for Climate Systems Research, Columbia University, New York, USA.
[7]Max-Planck-Institut für Meteorologie, Hamburg, Germany.
[8]University of Leeds, Leeds, United Kingdom.
[9]Department of Physics, Imperial College London, London, UK.
[10]Grantham Institute – Climate Change and the Environment, Imperial College London, London, UK.
[11]Norwegian Meteorological Institute, Oslo, Norway.
[12]NCAR/UCAR, Boulder, USA.
[13]Duke University, Durham, USA.
[14]University of Reading, Reading, UK.
[15]Department of Physics, University of Oxford, UK.
[16]Kyushu University, Fukuoka, Japan.

*Correspondence to*: Øivind Hodnebrog (oivind.hodnebrog@cicero.oslo.no)

**Abstract.**

Water vapour in the atmosphere is the source of a major climate feedback mechanism and potential increases in the availability of water vapour could have important consequences for mean and extreme precipitation. Future precipitation changes further depend on how the hydrological cycle responds to different drivers of climate change, such as greenhouse gases and aerosols. Currently, neither the total anthropogenic influence on the hydrological cycle, nor those from individual drivers, are constrained sufficiently to make solid projections. We investigate how integrated water vapour (IWV) responds to different drivers of climate change. Results from 11 global climate models have been used, based on simulations where $CO_2$, methane, solar irradiance, black carbon (BC), and sulphate have been perturbed separately. While the global-mean IWV is usually assumed to increase by ~7% per degree K surface temperature change, we find that the feedback response of IWV differs somewhat between drivers. Fast responses, which include the initial radiative effect and rapid adjustments to an external forcing, amplify these differences. The resulting net changes in IWV range from 6.4±0.9%/K for sulphate to 9.8±2%/K for BC. We further calculate the relationship between global changes in IWV and precipitation, which can be characterized by quantifying changes in atmospheric water vapour lifetime. Global climate models simulate a substantial

increase in the lifetime, from 8.2±0.5 to 9.9±0.7 days between 1986-2005 and 2081-2100 under a high emission scenario, and we discuss to what extent the water vapour lifetime provides additional information compared to analysis of IWV and precipitation separately. We conclude that water vapour lifetime changes are an important indicator of changes in precipitation patterns and that BC is particularly efficient in prolonging the mean time, and therefore likely the distance, between evaporation and precipitation.

## 1 Introduction

Water vapour is the largest contributor to the natural greenhouse effect and the source of a major climate feedback mechanism (Boucher et al., 2013). The global-mean integrated water vapour (IWV) is found to increase by around 7%/K both in models (Held and Soden, 2006; O'Gorman and Muller, 2010) and observations (Wentz et al., 2007; O'Gorman et al., 2012), consistent with the rate of change of saturation vapour pressure with temperatures representative of the lower troposphere and constant relative humidity (Allen and Ingram, 2002; Trenberth et al., 2003; Held and Soden, 2006). Hence, recent observed moistening trends have been attributed to human activities (Santer et al., 2007; Chung et al., 2014).

In contrast to the expected increase in IWV, models project that global-mean precipitation will only rise by 1-3% per degree of surface warming, due to energetic constraints (Allen and Ingram, 2002; Held and Soden, 2006; O'Gorman et al., 2012). Extreme precipitation events, however, are likely to increase with the availability of water vapour (Allen and Ingram, 2002) (at around 7%/K), but large uncertainties exist due to non-thermodynamic contributions (O'Gorman and Schneider, 2009; O'Gorman, 2015). Changes in the hydrological cycle will have widespread consequences for humanity, e.g., through changing precipitation patterns and extremes.

A number of recent studies have looked at the impacts of different climate drivers on the fast and slow components of the hydrological cycle separately, but most of these studies have focused mainly on precipitation (e.g., Andrews et al., 2010; Bala et al., 2010; Xie et al., 2013; Samset et al., 2016). Recently, new insight into precipitation changes has been given by analysing the atmospheric energy budget (Richardson et al., 2018b) and by the use of radiative kernels (Myhre et al., 2018a). In contrast to the slow (feedback) response, the fast response includes rapid adjustments to an external forcing and the initial radiative impact of the external forcing before changes in the global- and annual-mean surface temperature occur (Sherwood et al., 2015; Flaschner et al., 2016; Myhre et al., 2017). The common approach is to perform GCM simulations with prescribed sea surface temperatures (SST) to derive the fast response, and with coupled atmosphere-ocean to derive the total response. The slow response is the difference between the total and fast response.

The slow response in global precipitation scales with the surface temperature change induced by each driver (Andrews et al., 2010; Samset et al., 2016), while the fast response scales with the change in the atmospheric component of the radiative forcing. Black carbon (BC) differs from most other climate drivers due to strong regional solar absorption in the atmosphere, and has been identified as a driver with large inter-model variability (Stjern et al., 2017). Smith et al. (2018) explored rapid

adjustments due to different climate drivers, and found that changes in water vapour contribute a large part of these adjustments and oppose rapid adjustments due to tropospheric temperature changes.

The relationship between changes in IWV and precipitation (*P*) can be most easily examined by computing changes in atmospheric water vapour lifetime (*WVL*). The WVL then provides information on the extent to which this relationship is dependent on both the forcing mechanism and timescales of response, and the extent to which there is inter-model agreement on this relationship. The WVL is a fundamental component of the hydrological cycle and is useful for studying how dynamical processes in the hydrological cycle are altered due to climate change (Laderach and Sodemann, 2016), and is an important indicator of the transport length of water vapour (Singh et al., 2016).

The water vapour lifetime is also known as the residence time and is commonly expressed as the ratio between the time-averaged global-mean integrated water vapour and precipitation (Trenberth, 1998; Douville et al., 2002; Bosilovich et al., 2005; Schneider et al., 2010; Kvalevåg et al., 2013). The water vapour recycling rate is the inverse of the lifetime (P/IWV) and most often expressed regionally rather than globally (Li et al., 2011; Kao et al., 2018), in which another factor is how much of the regional precipitation results from transport of water vapour from outside the region. Studies identify a global-mean water vapour lifetime of 8-9 days for present-day conditions (van der Ent and Tuinenburg, 2017), although some argue that water only resides in the atmosphere for about 4-5 days (Laderach and Sodemann, 2016). A historical increase in WVL is found from both models (Bosilovich et al., 2005; Kao et al., 2018) and observations (Li et al., 2011; Kao et al., 2018). The fact that water vapour content increases more rapidly than precipitation with rising surface temperatures implies an expected increase in the lifetime (Douville et al., 2002; Held and Soden, 2006; Schneider et al., 2010).

Motivated by the great value of quantifying lifetimes of other quantities, e.g., BC aerosols (Bond et al., 2013), and the large number of studies that have focused on the topic of water vapour lifetime (e.g., van der Ent and Tuinenburg, 2017, and references therein), we want to explore historical and future changes in water vapour lifetime and discuss the potential value of quantifying WVL changes. In the first part of this study we use GCM results from the Precipitation Driver Response Model Intercomparison Project (PDRMIP) (Myhre et al., 2017) to explore how different climate drivers influence the distribution and magnitude of water vapour content throughout the atmosphere on both fast and slow timescales. In the second part, PDRMIP data are used to understand how the relationship between integrated water vapour and precipitation (i.e., the water vapour lifetime) has changed and is expected to change in the future according to GCM results in the Coupled Model Intercomparison Project phase 5 (CMIP5) (Taylor et al., 2011). We further discuss how changes in the WVL can be interpreted and the potential value of diagnosing the WVL in the context of global climate change.

## 2 Methods

### 2.1 Precipitation Driver Response Model Intercomparison Project (PDRMIP)

Data from 11 GCMs involved in PDRMIP have been used – details about PDRMIP and the participating models (except ECHAM-HAM – see Supplementary Text S1) are given in Myhre et al. (2017). The core PDRMIP experiments consist of

one base experiment, representing present-day conditions (pre-industrial for HadGEM2), and five perturbation experiments relative to base: doubling of the $CO_2$ concentration (hereafter denoted CO2x2), tripling of the $CH_4$ concentration (CH4x3), total solar irradiance increased by 2% (Sol+2%), five times increase in anthropogenic sulfate concentration or $SO_2$ emissions (SO4x5), and ten times increase in BC concentration or emissions (BCx10). Each experiment has been run with two model

set-ups: with fixed SSTs, and with a coupled model configuration, being run for at least 15 and 100 years, respectively. Analyses are here based on years 6-15 from the fixed SST experiments and years 51-100 from the coupled experiments. Each model has run one ensemble member, but the model-mean water vapour lifetime sensitivity (WVLS; see Section 2.3) for each experiment differs by only 3% or less if results from years 51-75 or 76-100 are used instead of years 51-100 from the coupled experiments; this indicates a strong signal-to-noise ratio.

All the PDRMIP models include aerosol-radiation interaction (direct aerosol effect) and associated cloud changes through changes in humidity and stability (rapid adjustments). Most models (except GISS-E2-R and NCAR-CESM1-CAM4) include aerosol-cloud interactions (indirect aerosol effects), which involve aerosol influences on cloud microphysics and are connected with large uncertainties (Boucher et al., 2013).

All model data have been regridded to T42 horizontal resolution, and, in the case of 3D data, to 60 vertical layers stretching

from the surface to 0.1 hPa. In Fig. S1, IWV in the PDRMIP base experiment has been compared with observations from MODIS Aqua and Terra level 3 data (downloaded from https://giovanni.gsfc.nasa.gov/giovanni/), and the cycle 36 output from the European Centre for Medium Range Weather Forecasts Integrated Forecast System model for year 2010. Four of the PDRMIP models did not have 3D fields with specific humidity available, but for these models the specific humidity was calculated based on temperature, pressure and relative humidity in each grid box and for each month.

**2.2 Coupled Model Intercomparison Project Phase 5 (CMIP5)**

Data from 26 GCMs participating in CMIP5 (Taylor et al., 2011) were obtained (see Fig. 4 for model names) for the historical (1850-2005) and RCP8.5 (a high emission pathway) (van Vuuren et al., 2011) (2006-2100) experiments, and for the variables surface air temperature, evaporation and water vapour path (here denoted integrated water vapour). The WVL was calculated by taking the global and 20-year mean IWV divided by evaporation (evaporation and precipitation are equal

in the global mean).

**2.3 Water vapour lifetime sensitivity**

The global-mean water vapour lifetime sensitivity follows the approach of Kvalevåg et al. (2013) and is calculated as

$$WVL_i = \frac{IWV_i}{P_i}$$

$$\Delta WVL = WVL_i - WVL_{base}$$

$$WVLS = \frac{\Delta WVL}{\Delta T_s}$$

where $WVL_i$ is the lifetime (in days), $IWV_i$ is the global-mean integrated water vapour (kg m$^{-2}$), and $P_i$ is the global-mean precipitation (kg m$^{-2}$ day$^{-1}$) for a perturbation experiment $i$. The water vapour lifetime change, $\Delta WVL$, is the difference between the lifetime in the perturbation and base experiments. The WVLS is the lifetime change divided by the global-mean surface temperature change, $\Delta T_s$. The $\Delta WVL$ due to fast responses has been split into contributions from $IWV$ and $P$ by

calculating the $\Delta WVL$ twice, with the $IWV$ and $P$ terms held constant one at a time. This assumption involves nonlinear terms, but the model-mean difference between the actual $\Delta WVL$ and the sum $\Delta WVL_{IWV} + \Delta WVL_P$ is less than 2% for all drivers.

## 3 Results and discussion

### 3.1 Zonal- and annual-mean changes in IWV

Table 1 shows that the slow responses of global-mean water vapour per degree K change in surface temperature are fairly close to the 7%/K that we expect from the Clausius-Clapeyron relation. However, the numbers differ somewhat between drivers, ranging from 6.5±1%/K for SO4x5 to 8.1±1%/K for Sol+2%. These differences become larger when the fast response is included, which adds to the pure surface temperature related response. The fast response is largest for BCx10, which changes from 7.5±1%/K to 9.8±2%/K between the slow and total response.

Integrated water vapour increases much more than evaporation and precipitation at nearly all latitudes and for all five PDRMIP drivers (Fig. 1; Fig. S2-3). However, the total global-mean increase in IWV differs strongly between each driver, with BCx10 at 9.8±2%/K and SO4x5 at 6.4±0.9%/K (Table 1). The estimated global IWV increase for BCx10 ranges from 6.8 to 13%/K for the different PDRMIP models, while locally decreasing in some regions (Fig. S4). BCx10, and to some extent SO4x5, show steep north-south gradients in the IWV change, emphasising the strong regional influences of these

short-lived compounds (Fig. 1; Fig. S3-4). The north-south gradient for BCx10 is steeper for the total response (Fig. 1b) than the slow response (Fig. 1a) due to strong influence of the fast response for this compound. In contrast to the other climate drivers, precipitation decreases and water vapour increases strongly for BCx10.

### 3.2 Changes to global-mean vertical profiles

In order to explore reasons for differences in IWV between the drivers, we compare vertical profiles of specific and relative

humidity and temperature for each of the climate drivers (Fig. 2). For the fast response from CO2x2, the change in the specific humidity profile differs considerably from what would be expected by the Clausius-Clapeyron relation when assuming that relative humidity stays constant (Fig. 2a; Fig. S5-6), a common assumption in climate change studies (Allen and Ingram, 2002). This indicates that the changes are not only temperature-driven. The specific humidity change for CO2x2 is around half of the expected temperature-induced change throughout most of the lower troposphere, explained by a

tropospheric relative humidity decrease that peaks near 800 hPa (Fig. 2c). This relative humidity decrease, and thus specific humidity decrease, contributes to the small fast IWV response for CO2x2 (Table 1). Over land, the lower than expected

increase in specific humidity is particularly evident in the lower troposphere (Fig. S5), and could be explained by the physiological effect since increased $CO_2$ leads to less evaporation from vegetation (Richardson et al., 2018a). CH4x3 shows some of the same tendency as CO2x2 (Fig. 2a) but without any considerable change in relative humidity (Fig. 2c). Sol+2% largely follows the temperature-induced change in specific humidity (Fig. 2a).

In contrast to the CO2x2 experiment, BCx10 mostly yields a small increase in relative humidity (Fig. 2c), especially close to the surface, and therefore the specific humidity change for BCx10 is larger than the temperature-induced change throughout most of the troposphere (Fig. 2a). This low-level relative humidity increase contributes to the large fast IWV response for BCx10 (Table 1). Additional contributions come from atmospheric solar absorption due to BC, which leads to rapid atmospheric temperature increase (Fig. 2d) and therefore increased water vapour availability. It is also worth noting the

different lapse rates between the drivers, broadly with temperature changes decreasing with height for CO2x2 and increasing with height for BCx10 (Fig. 2d; Fig. S6), and this has implications for atmospheric stability.

Changes in specific humidity profiles for the slow response (Fig. 2b) show that the assumption of constant relative humidity does hold. When normalized with $\Delta T_s$, the specific humidity profiles are similar between the drivers, with a small exception for SO4x5, which shows a smaller increase throughout the troposphere. One reason is that SO4x5 is the driver that gives the

least change in the temperature profile (and lapse rate up to 300 hPa) when normalized with surface temperature change (Fig. S7, lower left), and therefore the least change in the water vapour availability. It is also worth noting the strong difference between land and sea in the temperature change profile for this driver. The small increase in vertical temperature profiles compared to the other drivers could explain why SO4x5 has the smallest slow IWV response (Table 1). Sol+2%, which have the largest slow IWV response, has the second strongest increase in temperature profile per K surface temperature change

(Fig. S7). BCx10 gives the strongest increase in atmospheric temperature, but a decrease in relative humidity, especially over land (Fig. S7), leads to a discrepancy between the actual specific humidity change and the temperature-driven change between the surface and 800 hPa (Fig. 2b).

Earlier studies have shown a strong land-ocean contrast in the response of near-surface relative humidity to global warming, mainly due to greater warming over land than ocean (Byrne and O'Gorman, 2016). Inspection of near-surface relative

humidity changes shows that patterns of reduced relative humidity over land and increased over oceans are rather similar between drivers for the slow response (Fig. 3). However, the fast response constitutes a large part of the total response for all drivers. For CO2x2, fast responses amplify the land-ocean contrast considerably, while for BCx10, fast responses lead to strong increases in relative humidity over large land regions, and these outweigh the reductions over land in the slow response. Interestingly, Sol+2% shows much less land-ocean contrast than CO2x2 in the fast response, while their slow

responses are very similar.

## 3.3 Water vapour lifetime and sensitivity

The CMIP5 pre-industrial multi-model mean value for the WVL is 7.8±0.5 days (Fig. 4a). All models show an increase over both the historical and future time period (a paired sample *t*-test shows that the multi-model mean increases are significant).

A substantial increase of the lifetime from a present-day (i.e. 1986-2005) value of 8.2±0.5 to 9.9±0.7 days towards the end of the century is projected by the mean of CMIP5 models assuming RCP8.5, because increases in IWV are larger than for precipitation (Fig. 4b). Also, nearly 75% of the models show a stronger WVLS for the historical period than for the future, with model-mean values of 0.55±0.1 days/K and 0.47±0.06 days/K, respectively, and a paired sample $t$-test shows that the two values are significantly different. The present-day lifetime of 8.2±0.5 days from CMIP5 is close to, but slightly lower than, a recent assessment using reanalysis data of 8.9±0.4 days (van der Ent and Tuinenburg, 2017).

To further understand these differences, it is instructive to investigate WVLS for each of the fast and slow responses, and for each of the five climate drivers studied in PDRMIP. The response to surface temperature changes (i.e., slow response) dominates the change for all drivers except BCx10 (Fig. 5a). However, the fast response is still a significant enhancement to the slow response for CO2x2 (27% of the total). For CH4x3, the fast response is 30% of the total, but with large differences between models (range of 8%-58%). All models (except HadGEM3) show that the fast response is more important than the slow response for BCx10, because BC is the driver with the strongest atmospheric temperature increase for the fast response. These results support earlier single-model findings (Kvalevåg et al., 2013). The slow response is remarkably similar between models and drivers, again with the exception of BCx10, which has an inter-model range of 0.10-0.45 days/K. The total WVLS is more than twice as large for BCx10 than for any other driver, and this can be explained by the strong increase in integrated water vapour combined with a decrease in precipitation, in contrast to the other climate drivers which enhance precipitation (Fig. 1b).

Separating the fast response into contributions from changes in atmospheric water vapour and precipitation (keeping in mind that the lifetime is defined as global water vapour divided by precipitation) reveals that both terms are large for BCx10, but that reduced precipitation dominates the fast WVL changes (Fig. 5b). Interestingly, reduced precipitation also dominates the fast WVL changes for CO2x2, while increased atmospheric water vapour dominates for Sol+2%. For SO4x5, the small fast WVL change is dominated by reduced water vapour (note that increased sulphate leads to cooling) except for one model (NCAR-CESM1-CAM5), which has a different sign because it has perturbed emissions rather than concentrations and it includes the influence of sulphate on BC through coating. This leads to a heating of the atmosphere, and this effect dominates the direct sulphate effect because fast responses for sulphate are small.

## 3.4 Historical lifetime changes explained

By combining the PDRMIP results for individual drivers with radiative forcing since pre-industrial time, we can reproduce the pre-industrial to present-day WVL increase of 0.34±0.08 days from CMIP5 models within the uncertainties (Fig. 6a). There is an almost equal contribution from the slow temperature response and the fast response to the total lifetime change. The PDRMIP estimate of historical lifetime change due to the slow (temperature) response in Fig. 6a was derived by first taking the mean of the slow lifetime change across all PDRMIP drivers in Fig. 5a. This value of 0.31 days K$^{-1}$ was then multiplied with the multi-model mean CMIP5 historical surface temperature change of 0.64 K (not shown). The PDRMIP fast response contribution in Fig. 6a is the sum of the individual terms in Fig. 6b. These terms have been derived by

combining the present-day radiative forcing for separate climate drivers from Myhre et al. (2013) with the radiative forcing and fast WVL change from PDRMIP models using the following equation for each PDRMIP model

$$\Delta WVL_{fast,historical,j} = \Delta WVL_{fast,PDRMIP,j} \times \frac{RF_{historical,j}}{RF_{PDRMIP,j}}$$

where $RF$ is the radiative forcing (in W m$^{-2}$) and $j$ designates the climate driver and corresponding PDRMIP experiment, e.g.,

$CO_2$ and CO2x2, respectively (see Table S1 for details and multi-model mean values). By scaling the fast $\Delta WVL$ with RF, we are able to estimate the historical contribution to $\Delta WVL$ from each driver, in a similar way to what has been done before for other quantities (e.g., sensible heat flux changes in Myhre et al. (2018b)). The bars in Fig. 6b have further been split into contributions from changes in precipitation and water vapour using the numbers in Fig. 5b. All calculations were done for each PDRMIP model, and Fig. 6 shows the multi-model mean results.

Disentangling the fast response into contributions from the main historical climate drivers shows that increased $CO_2$ concentrations constitute around half (~0.1 days) of the net increase due to the fast response, with the reduced precipitation term being three times as large as the contribution from increased water vapour (Fig. 6b). For aerosols, a substantial lifetime increase due to BC is partly counteracted by scattering aerosols, which reduce the WVL. The impact of aerosols contributes to the stronger WVLS in the historical period compared to the future simulations in the CMIP5 models (Fig. 4b), since

aerosol and aerosol precursor emissions are projected to decrease strongly towards the end of the century in the RCPs (Rogelj et al., 2014).

### 3.5 Interpretation and value of water vapour lifetime

In atmospheric chemistry and aerosol science, the lifetime of a compound is a measure of how long the compound resides in the atmosphere after it is emitted or produced, e.g., through surface emissions or chemical production. Diagnosing this

lifetime has proved useful in many aspects, and the most relevant example is probably the lifetime of BC. Several studies show that the BC lifetime in many global aerosol models is too long, and reducing the lifetime by altering the sinks (mainly wet removal for BC) has given substantial improvements in modelled vertical profiles compared with observations, with large implications for the climate effect of BC (e.g., Hodnebrog et al., 2014; Samset et al., 2014). In a similar way to chemical compounds, the changes in atmospheric water vapour lifetime (WVL) is the relationship between changes in the

burden (i.e., IWV) and the sources/sinks (i.e., evaporation/precipitation). In the following we discuss the potential value of diagnosing changes in the water vapour lifetime in addition to examining changes in IWV and P separately.

Since the water vapour lifetime includes both IWV and P, it can be used to measure variations in the hydrological cycle and be an important indicator of climate change (Kao et al., 2018). However, the interpretation of changes in the hydrological cycle can be rather confusing and deserves some discussion. While the longer WVL induced by global warming means that

the hydrological cycle is slowing down, the global-mean precipitation or evaporation fluxes are also commonly referred to as the strength of the hydrological cycle; because they both increase, this implies an intensification or acceleration of the hydrological cycle with global warming (e.g., Wu et al., 2013). Hence, when the global hydrological cycle is said to intensify

or accelerate with warming, it should be made clear that this refers to the fluxes and not the cycle as a whole. Hence, as noted by Douville et al. (2002), the conclusion that the hydrological cycle is intensifying is somewhat misleading because it suggests faster turnover of water, which is not the case; instead they use the term amplification to indicate an increase in precipitation and evaporation rather than acceleration (which implies a decreased lifetime) of the hydrological cycle. While the terminology could be confusing, both the amplification (through intensification of fluxes) and the slowdown (through longer lifetime) are important indicators of changes in the hydrological cycle. The intensification of fluxes means more precipitation globally and higher water availability, with potential consequences for extreme precipitation and water vapour feedback. The slowdown, however, is an important sign of changing precipitation patterns, since water vapour resides in the atmosphere for a longer time before precipitation, and this behaviour cannot be deduced based on analysing IWV and P separately. A longer water vapour lifetime implies an increased length scale of water vapour transport, so that the distance between evaporation and precipitation of moisture is greater, as has been shown in detail by Singh et al. (2016). They further mention implications of this increased transport length scale, such as the expansion of the Hadley circulation and a poleward shift of midlatitude precipitation maximum.

Another aspect of WVL is the link to isotopes. Stable water isotopes provide valuable knowledge on the evaporation and condensation history of atmospheric moisture, and more specifically on, e.g., proportions of convective and stratiform precipitation (Aggarwal et al., 2016) and past variability in high-latitude aerosol abundance (Markle et al., 2018). Singh et al. (2016) highlight, due to changes in water vapour lifetime and transport length scale, that caution is needed when interpreting isotope data. Also, a positive correlation between WVL and stable isotope ratio in precipitation has been found from daily measurements at stations representing a range of climate regimes (Aggarwal et al., 2012), and better diagnostics of the impact of WVL on isotopes have been called for (Dee et al., 2018). It is suggested that the relationship between isotope ratio and the WVL could be used to improve the parameterizations of vertical mass-exchange in global climate models (GCMs) (Aggarwal et al., 2012), which is currently one of the major uncertainties in GCMs (Bony et al., 2015). Therefore, understanding the WVL has the potential to contribute to improved quantification of the hydrological cycle and its climate-induced changes.

Among the most important caveats with our WVL findings is that climate models have known deficiencies, such as problems with representing vertical convective mass fluxes (Bony et al., 2015), surface moisture fluxes and entrainment/detrainment rates. Part of the reason is that GCMs have relatively coarse resolution and many processes, such as convection, need to be parameterized. The model spread in future WVL change is lower (relative standard deviation (RSD) of 22%) than for, e.g., precipitation change (RSD of 30%) (Fig. 4), but the horizontal resolutions in the PDRMIP models range from 1.4°×1.4° (MIROC-SPRINTARS) to 2.8°×2.8° (CanESM2), where convection needs to be parameterized. This means that the uncertainty is larger if a decrease in convective mass fluxes is a major reason for the increase in WVL. Compared with present-day WVL from reanalysis, the climate models have too short WVLs (Trenberth et al., 2011; see also Section 3.3). Kao et al. (2018) compared trends in precipitation and column water vapour data from 13 CMIP5 models with observational datasets and also found differences in the moisture recycling rate between observations and the CMIP5

models, and concluded that this discrepancy was caused by relatively poor simulations of precipitation. However, the long-term trend and inter-annual variability of column water vapour was very well captured by nearly all models.

In conclusion, diagnosing changes in the water vapour lifetime reveals important changes in the hydrological cycle, and specifically changes in precipitation patterns since these are directly affected by how long water resides in the atmosphere. Since black carbon is the climate driver among those studied here with the by far strongest water vapour lifetime sensitivity, it is also the driver that is most efficient in increasing the distance between evaporation and precipitation per degree K global surface temperature change. However, the full potential of diagnosing the water vapour lifetime is most likely still unexplored. When inclusion of isotopes in GCMs becomes more common in the future, WVL and its changes can potentially prove extremely useful in constraining projections of the hydrological cycle and precipitation, in a similar way to how diagnosing and evaluating the lifetime of BC has helped constrain the climate effect of BC.

## 4 Conclusions

Based on new model simulation data we have investigated how different climate drivers influence water vapour in the atmosphere. We find that the feedback response of IWV, the relative change per degree K global- and annual-mean surface temperature change, differs somewhat between drivers, ranging from 6.5%/K for sulphate to 8.1%/K for solar forcing. Fast responses are particularly important for black carbon because of rapid heating of the atmospheric column, leading to an increase from 7.5 to 9.8%/K between the feedback response and the total IWV response, and with strong regional differences in the IWV distribution. For $CO_2$, fast responses are also important, leading to a decrease from 7.7 to 7.2%/K, for the slow and total response respectively, partly due to a reduction of relative humidity throughout the troposphere in the fast response. We also show that the fast response is an important contributor to the previously known strong land-ocean contrast in the response of near-surface relative humidity to global warming, with $CO_2$ and BC showing strong and opposite fast responses over land.

Results show that the lifetime of water vapour could increase by 25% by the end of the 21[st] century in a high emission scenario. This is because of the large expected temperature changes, and despite the projected aerosol emission reductions leading to a lower water vapour lifetime sensitivity. Among the climate drivers studied here ($CO_2$, methane, solar irradiance, BC, and sulphate), WVL changes are most sensitive to perturbations in BC aerosols (1.1±0.4 days per Kelvin increase in $T_s$), due to strong increases in IWV with temperature combined with a precipitation reduction (in contrast to a positive precipitation change per unit temperature change for other drivers). According to model calculations, an increase in WVL of 4-5% between pre-industrial and present-day has already occurred, and around half of this increase is due to fast atmospheric responses. Aerosol concentration changes, and BC in particular, strongly modify the fast WVL change and contribute to large inter-model uncertainty.

The increase in WVL with global warming reveal important changes in the hydrological cycle. Quantifying WVL changes gives information about changing precipitation patterns – information that cannot be deduced by analysing IWV and P

separately. More specifically, a longer lifetime leads to greater distances between the source (evaporation) and sink (precipitation) of water vapour, with implications such as the Hadley cell expansion (Singh et al., 2016). Our results show that BC is considerably more efficient than any of the other climate drivers in prolonging the WVL, and therefore likely the transport length of water vapour. Estimating WVL could become more important in the future as inclusion of isotopes in GCMs becomes more common, and this may lead to more robust projections of the hydrological cycle and precipitation.

## Data availability

The PDRMIP model results are available at http://cicero.uio.no/en/PDRMIP. The CMIP5 data are available at http://pcmdi9.llnl.gov/.

## Author contribution

GM and ØH designed the study. ØH performed the analysis and was the primary writer and received input from all authors, especially GM.

## Competing interests

The authors declare that they have no conflict of interest.

## Acknowledgements

Ø.H., G.M., B.H.S. and K.A. were funded by the Research Council of Norway (RCN), through the grant NAPEX (229778). Supercomputer facilities were provided by NOTUR. Simulations with GISS-E2 used resources provided by the NASA High-End Computing Program through the NASA Center for Climate Simulation at Goddard Space Flight Center. M.K. and A.V. were supported by the Natural Environment Research Council (NERC) under grant number NE/K500872/1. Simulations with HadGEM3-GA4 were performed using the MONSooN system, a collaborative facility supplied under the Joint Weather and Climate Research Programme, which is a strategic partnership between the Met Office and NERC. T. T. was supported by the supercomputer system of the National Institute for Environmental Studies, Japan, the Environment Research and Technology Development Fund (S-12-3) of the Ministry of the Environment, Japan and JSPS KAKENHI Grant Number JP15H01728 and JP15K12190. D.O. and A.K. were supported by RCN through the projects EarthClim (207711/E10), EVA (229771), the NOTUR (nn2345k) and NorStore (ns2345k) projects, and through the Nordic Centre of Excellence eSTICC (57001) and the EU H2020 project CRESCENDO (641816). T.B.R. was supported by NERC training award NE/K007483/1, and acknowledges use of the MONSooN system. Computing resources for J.F.L. (ark:/85065/d7wd3xhc) were provided by the Climate Simulation Laboratory at NCAR's Computational and Information Systems Laboratory, sponsored by the

National Science Foundation and other agencies. Computing resources for the simulations with the MPI330 ESM model were provided by the German Climate Computing Center (DKRZ), Hamburg. Computing resources for O.B. were provided by GENCI at the TGCC under allocation gen2201. T.A. was supported by the Joint UK BEIS/Defra Met Office Hadley Centre Climate Programme (GA01101). P.S. acknowledges funding from the European Research Council project RECAP under the EU's H2020 research and innovation programme with grant agreement 724602 and the EU's FP7/2007-2013 projects BACCHUS under grant agreement 603445. D.W.P. acknowledges funding from NERC projects NE/L01355X/1 (CLARIFY) and NE/J022624/1 (GASSP). The ECHAM-HAM simulations were performed using the ARCHER UK National Supercomputing Service. We acknowledge the World Climate Research Programme's Working Group on Coupled Modelling, which is responsible for CMIP, and we thank the climate modeling groups for producing and making available their model output. For CMIP the U.S. Department of Energy's Program for Climate Model Diagnosis and Intercomparison provides coordinating support and led development of software infrastructure in partnership with the Global Organization for Earth System Science Portals. We thank Adriana Bailey for comments and Camilla Stjern for comments to an earlier version of the manuscript. We further thank three anonymous reviewers for valuable comments.

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

**Table 1. Global- and annual mean relative changes in integrated water vapour and with evaporation/precipitation in parentheses (note that global-mean evaporation and precipitation are equal) for five different drivers, and split into fast, slow and total responses, using the mean of PDRMIP results. In the two rightmost columns, values have been divided by the global- and annual-mean $\Delta T_s$ induced by each driver.**

|        | Fast (%)    | Slow (%)     | Total (%)    | Slow (%/K) | Total (%/K) |
|--------|-------------|--------------|--------------|------------|-------------|
| CO2x2  | 0.8 (-2.5)  | 16.8 (6.1)   | 17.6 (3.6)   | 7.7 (2.8)  | 7.2 (1.4)   |
| CH4x3  | 0.4 (-0.5)  | 4.6 (1.9)    | 5.0 (1.4)    | 7.4 (3.1)  | 7.3 (1.8)   |
| Sol+2% | 1.2 (-0.7)  | 18.7 (6.7)   | 19.9 (6.0)   | 8.1 (3.0)  | 8.2 (2.5)   |
| BCx10  | 2.1 (-3.0)  | 4.4 (1.5)    | 6.5 (-1.5)   | 7.5 (2.8)  | 9.8 (-3.4)  |
| SO4x5  | -0.4 (-0.1) | -13.3 (-6.1) | -13.7 (-6.1) | 6.5 (2.9)  | 6.4 (2.8)   |

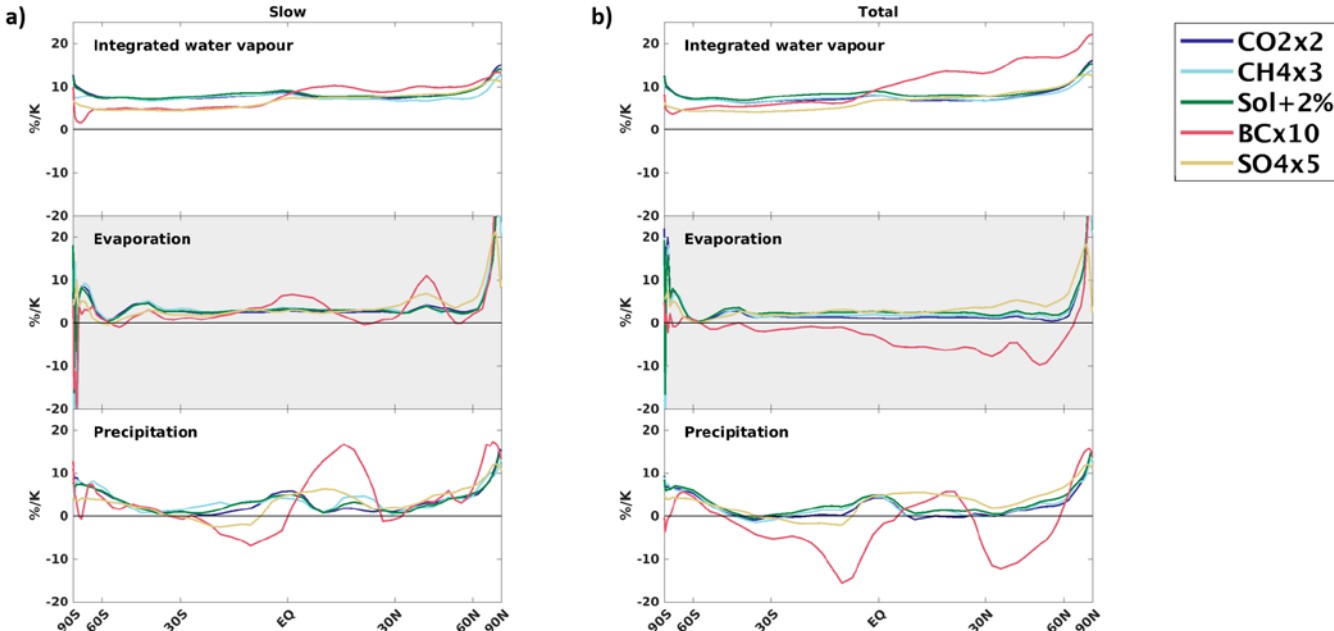

**Figure 1: Zonal-mean relative changes (in %/K) in integrated water vapour, evaporation, and precipitation for five different drivers for the a) slow and b) total response, divided by the global- and annual-mean $\Delta T_s$ induced by each driver, using the mean of the PDRMIP results. In some cases, relative evaporation changes are large at very high latitudes and exceed the scale.**

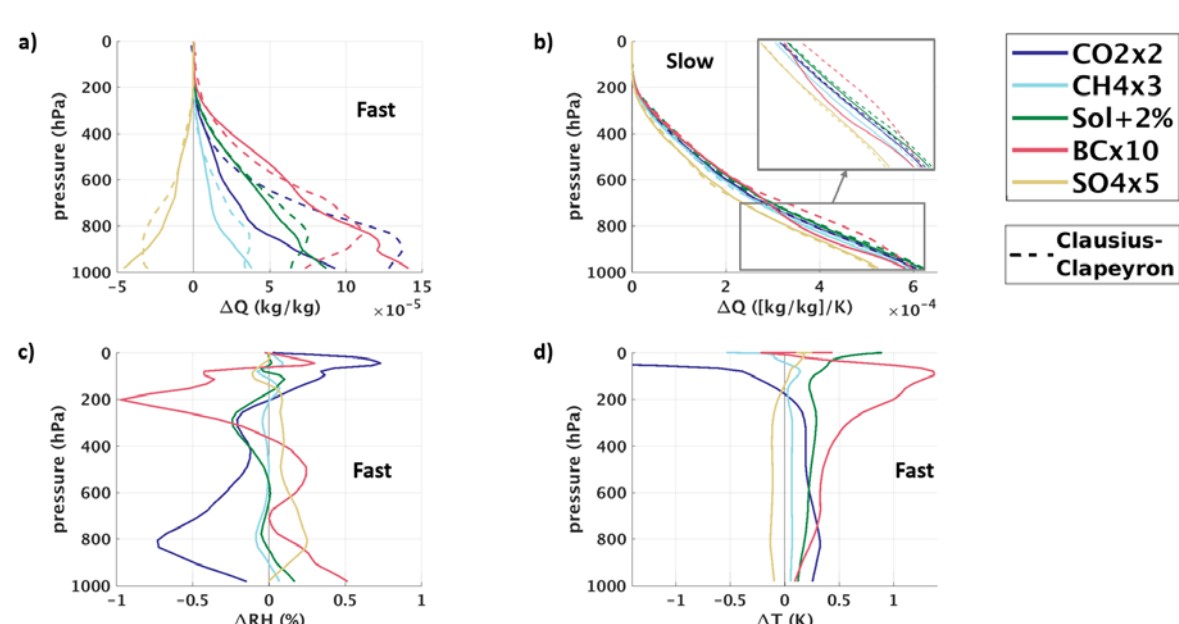

**Figure 2: Vertical profile changes for individual drivers. a) Fast and b) slow changes in specific humidity ($\Delta Q$), and fast changes in c) relative humidity ($\Delta RH$) and d) temperature ($\Delta T$), using the mean of the PDRMIP results. In a) and b), dashed lines show expected specific humidity changes from the Clausius-Clapeyron relation assuming constant relative humidity (calculated for each**
10 **model, month and grid box, and with values at pressures <10 hPa set to zero because this approximation does not hold for low pressures). The slow response in b) is divided by $\Delta T_s$ induced by each driver, and the inset plot is zoomed in on 1000-700 hPa.**

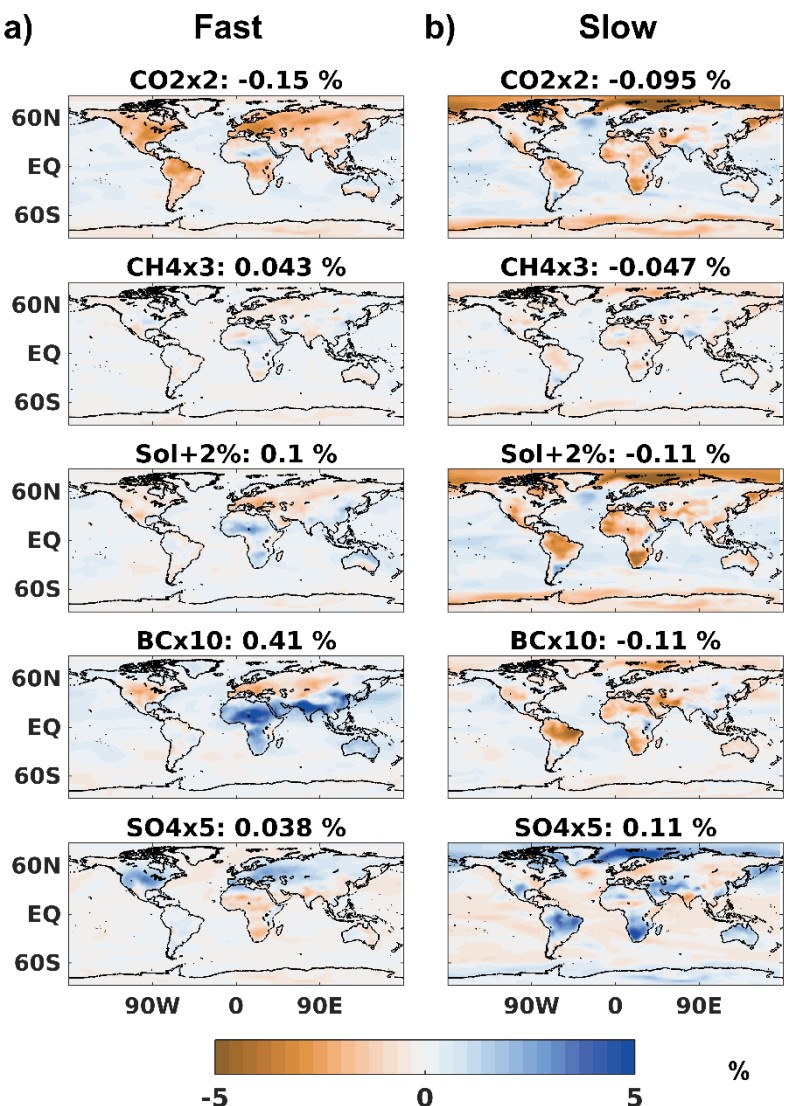

**Figure 3: Maps of model mean absolute change in near-surface relative humidity (%) for each PDRMIP driver, separated into a) fast and b) slow responses. The plots are means of the six models with available data for near-surface relative humidity: CanESM2, HadGEM2, MIROC-SPRINTARS, NCAR-CESM1-CAM4, NCAR-CESM1-CAM5 and NorESM1.**

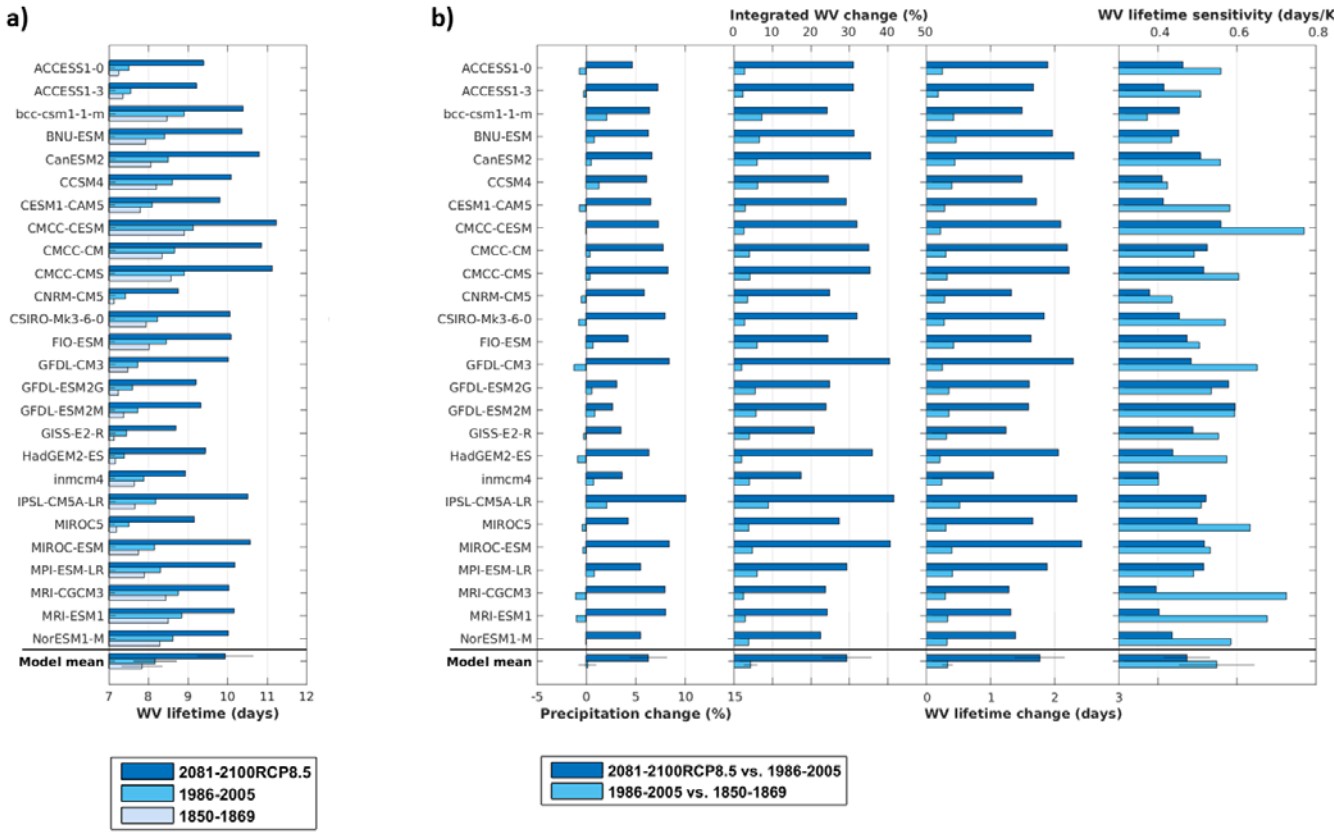

**Figure 4: Historical and future water vapour lifetime in CMIP5 models. a) Water vapour (WV) lifetime (in days) for each of the three time periods, and b) changes in precipitation (%), integrated WV (%), WV lifetime (days), and WV lifetime sensitivity (WVLS; days/K) between each of the time periods. Error bars show the standard deviation representing the spread between the models. All values are global- and annual-means.**

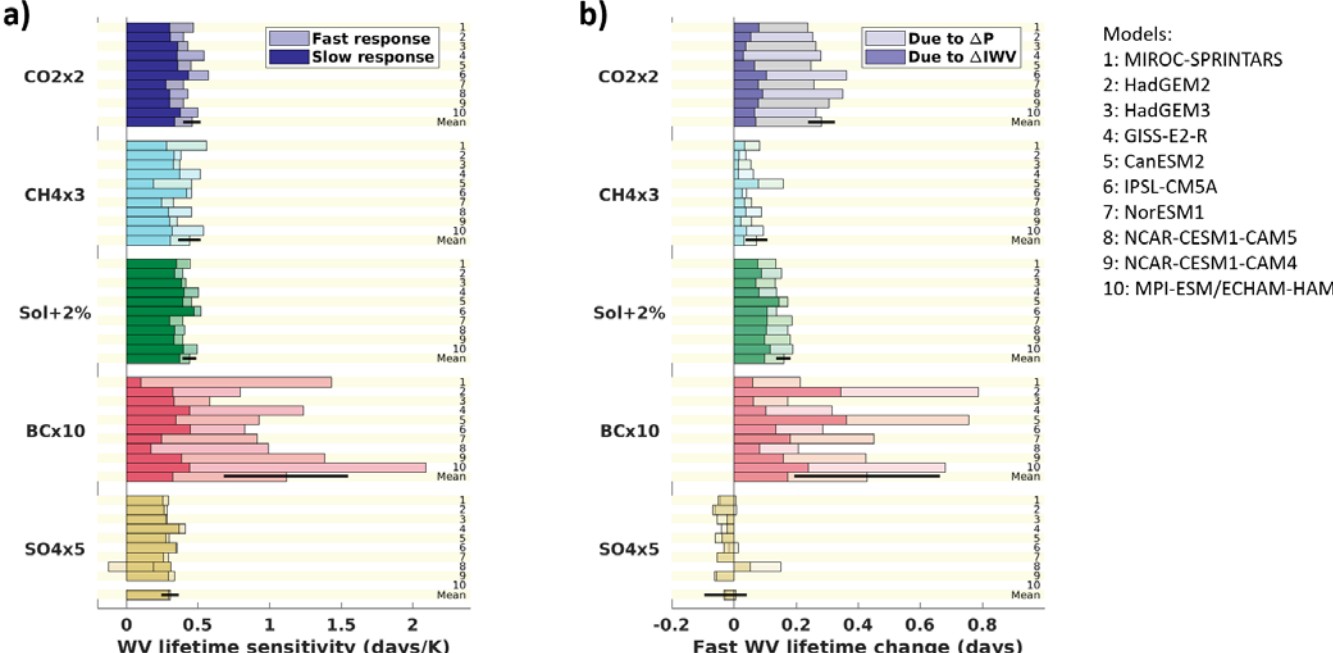

**Figure 5: Water vapour lifetime changes for individual drivers. a)** Water vapour (WV) lifetime sensitivity (in days/K) in PDRMIP models, split into slow (dark-coloured bars) and fast (light-coloured bars) responses for each driver. **b)** WV lifetime change (days) due to fast responses, split into contributions from changes in atmospheric water vapour (dark-coloured bars) and precipitation (light-coloured bars). The fast response in a) is not divided by $\Delta Ts$ but calculated as the difference between the total and slow WV lifetime sensitivity (in days/K). In the few cases where the dark and light-coloured bars have opposite sign (e.g., SO4x5 model no. 8 in Fig. 5a), the vertical black line gives the net value. Error bars show the standard deviation representing the spread between the models.

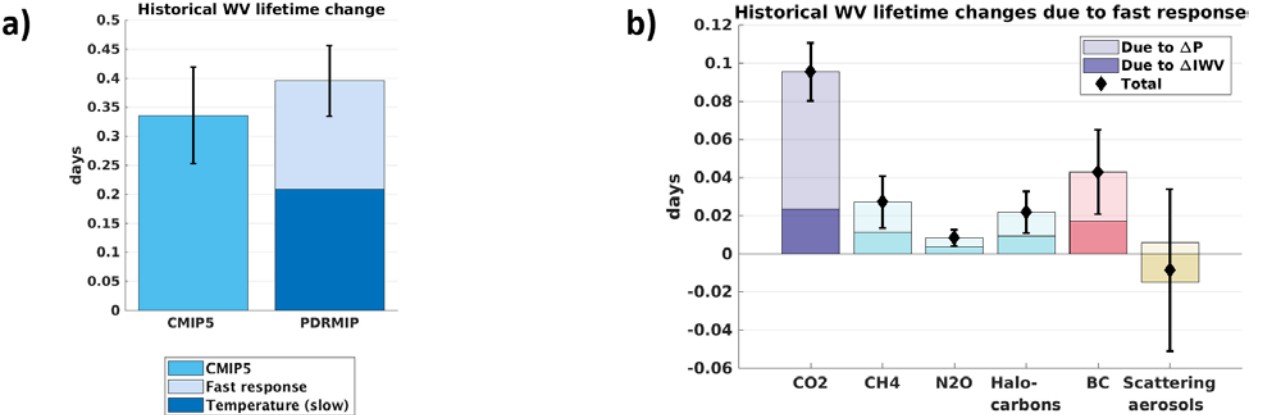

**Figure 6: Contributions to historical water vapour lifetime change. a)** Total historical change in water vapour (WV) lifetime from CMIP5 models compared to PDRMIP results with contributions from slow and fast responses. **b)** Historical WV lifetime change due to fast responses, split into different drivers and into contributions from changes in precipitation (light-coloured bars) and IWV (dark-coloured bars). Error bars show the standard deviation representing the spread between the models.