# Peer review of "Water vapour adjustments and responses differ between climate drivers"

_Atmospheric Chemistry and Physics, 2019_

## Referee Comment (RC1) · Anonymous Referee #2 · 15 Apr 2019

A catalog of idealized climate model experiments to assess precipitation response to different radiative forcings is here exploited to investigate how the atmospheric lifetime of water vapor is affected. This is diagnosed as integrated water vapor divided by precipitation rate which effectively characterizes how long it would take to precipitate out all the water vapor in the atmospheric column. Although it is obvious, based on past research, that this lifetime should increase, since thermodynamic and energetic constraints cause water vapor to increase at a faster fractional rate than precipitation, this work provides a useful investigation into the differences in this response between forcing agents, relating to fast adjustments and slow response to temperature, and further explores regional contributions. The most novel aspect, in my view, may be demonstrating how water vapor adjustments and responses differ between forcing agents. I

recommend emphasizing this and I consider that this work merits publication follow-ing consideration of the suggestions below including the possibility of comparing with observed responses.

Specific points

1) p.1, L24-26: the first 2 lines do not make much sense to me in the abstract and have marginal relevance to the results. Something outlining what water vapor lifetime is and why it is important would be more useful I think.

2) p.1, L29: "projected" –> "simulated" (1986-2005 is not a projection)

3) p.1, L31: "slows down the hydrological cycle" - if precipitation is increasing, the hydrological cycle could be thought of as speeding up since water is fluxing between atmosphere and surface more quickly so I suggest removing this confusing terminol-ogy.

4) p.1, L34 - "fast responses" should be clarified

5) p.2, L18-20 - Is there a difference between water vapor residence time, lifetime and recycling rate (e.g. Li et al. 2011; Kao et al. 2018; van der Ent & Tuinenburg (2017); Allan & Zveryaev (2011) IJOC http://doi.org/10.1002/joc.2070). This could be clarified. Regional responses in water vapor lifetime may be misleading since the precipitation can result from transport of moisture from outside of the region and so not really reflect recycling rate within a box

6) p.4 L26 - RCP8.5 is a high emissions scenario but cannot simply be described as a business as usual pathway.

7) p.5, L15 - is an increase in WVL detectable in the historical period 1986-2005? Using trends from Allan et al. (2014) Surv. Geophys. http://doi.org/10.1007/s10712-012-9213-z for 1988-2008 and assuming WVL=8.9 days:

WVLS = WVL((1/IWV)(dIWV/dT) - (1/P)(dP/dT)) = 8.9x(0.064-0.028) = 0.32 days/K,

which is smaller than simulated perhaps due to additional noise from internal variability (with a large uncertainty). Alternatively:

dWVL/dt=WVL((1/IWV)(dIWV/dt) - (1/P)(dP/dT)) = 8.9x(0.0084 - 0.0018) = 0.06 days/decade (rather small)

8) Fig.2 - additional annotation to show the meaning of dark/light bars in (a) and (b) would help the reader.

9) Fig.3 - it is not clear from the scattering aerosol bar how the light and dark part contribute. Perhaps the total can be distinguished as a thick horizontal line or symbol (at the top of most bars but at -0.01 for scattering aerosol).

10) Fig. 4 - it would be more informative for me to group all the WV, E and P lines into 3 separate plots so that they can be compared across forcing agents. Are zonal values calculated using zonal dT or global dT?

11) p.7, L4-6 seems an important result and some more mechanistic discussion of this would be useful. Is the SO4 slow response small due to forcing predominantly affecting land which has less moisture availability? Or does this relate to the vertical temperature changes and the temperature dependence of the Clausius Clapeyron equation? Does the low level relative humidity increase explain the large fast response in BCx10 and why? On the other hand is this all explained by land-ocean temperature responses as implied? This could be summarized in the conclusions along with implications (why do we care?).

12) p.7, L26 "small exception for SO4x5." Please be more explicit in what is meant.

13) P.7, L30 - there is very little mention of Figure 6. Either this can be removed or a little more discussion of the Figure panels included.

14) Fig.5 - dashed=Clausius Clapeyron in the legend would help. It is difficult to see dashed and solid in (b) so perhaps this can be replaced with a relative humidity change plot.

15) p.8 (Conclusions) - what is the significance of changes in water vapor lifetime above the differing fractional responses of P and IWV and implications for changes in the tropical circulation mass flux and precipitation intensity distribution, which is well known? Emphasizing what is novel will help increase the impact of this work.

---

## Referee Comment (RC2) · Anonymous Referee #3 · 30 Apr 2019

A number of climate model simulations from the CMIP5 intercomparison are used in order to estimate the change in water vapor lifetime with climate change. Water vapor lifetime is shown to increase by about 2 days in the next 100 years. Contributions from different climate drivers are analyzed using simulations from the Precipitation Driver Response Model Intercomparison Project (PDRMIP). Estimates for the combination of all drivers for the past are shown to be consistent with CMIP5 results. Changes in WVL are split into fast and slow responses. Changes in IWV per surface temperature change of different climate drivers are compared to the theoretical 7%/K increase that is expected assuming relative humidity to stay constant. BC shows the strongest increase in water vapor lifetime. The findings are very interesting but the paper is too concise to appreciate results fully. More information, explanations for assumptions and discussion

needs to be added.

1. You calculate contributions from changes in IWV and P to ΔWVL by calculating the ΔWVL twice, with the IWV and P terms held constant one at a time (page 5 line 9-10). This means that you neglect nonlinear terms which needs to be mentioned. It is difficult to judge from the material presented if this is a good assumption, since figure 2a gives the fast WVLS and figure 2b the WVL itself. I suggest plotting the overall WVL change in figure 2b additionally.

2. Could you please give an explanation why it makes sense to scale ΔWVL with RF (page 6 line 17).

3. Water vapor lifetime is increased which is supposed to be connected with a decrease in vertical mass fluxes. But a decrease in vertical mass fluxes should be connected with a moistening of the lower troposphere which appears not to be the case. Is there an explanation for this behavior?

4. Changes of water vapor lifetimes are connected with vertical mass fluxes. For the analysis of WVL changes you use climate models which have problems representing those mass fluxes. In particular convective mass fluxes are known to be a source of large uncertainty within climate models. Surface moisture fluxes may also be problematic. Vertical profiles of humidity may be strongly dependent on entrainment and detrainment rates which are highly problematic. It would be good to add a discussion about how dependent results are on known deficiencies in global models. Original model resolutions need to be given.
* * *

---

## Referee Comment (RC3) · Anonymous Referee #1 · 6 May 2019

The authors present a study of water vapour lifetime, defined as the ratio of IVW/P, from a series climate models for present and future climate. The main driving forces of an increase of the lifetime in a warmer climate are analysed by a range of sensitivity studies. In addition to some major comments, the overall writing can be substantially improved in some sections. I therefore recommend major revisions.

**Major comments**

1. Abstract: The abstract should be revised to better reflect the actual content of the manuscript.

2. Introduction: "The fact that water vapour content increases more than precipitation with rising surface temperatures implies an expected increase in the lifetime

(Douville et al., 2002; Held and Soden, 2006; Schneider et al., 2010), and hence a slowing down of the hydrological cycle. However, global-mean precipitation or evaporation fluxes are commonly referred to as the strength of the hydrological cycle, which, in contrast, implies an intensification or acceleration of the hydrological cycle with global warming. Douville et al. (2002) note that this conclusion is somewhat misleading because it suggests faster turnover of water, which is not the case. Hence, when the global hydrological cycle is said to intensify or accelerate with warming, it should be made clear that this refers to the fluxes and not the cycle as a whole"

The takeaway from this paragraph is rather confusing. If the fluxes intensify, what is the significance of the overall slowdown? The authors need to resolve the more fundamental underlying issue of explaining the meaning or significance of a residence time change. If what matters for the impact of climate change is the intensification of fluxes, what is the purpose of talking about a slowdown of the hydrological cycle? Maybe there is a clear answer, but it needs to be stated somewhere early on to motivate the reader to adopt the perspective of IWV/P rather than IWV and P individually.

3. Regarding the deep convective mass flux in the tropics: If the motivation of the study is to compare the convective mass flux between the different models, it may be relevant to consider additional quantities, such as convective vs. stratiform precipitation (noting that convective precipitation parameterisations have a large uncertainty and differ substantially between climate models), or the mass flux itself. As the study is designed presently, the mass flux and the implications thereof are rather implicit, and it remains unclear whether this quantity should be considered as an internal model variable or as the actual flux of mass as represented by climate models.

4. On pg. 3, L. 10 onward, the authors state that "understanding the WVL has the potential to contribute to improved quantification of the hydrological cycle and its

climate-induced changes", based on a previous paragraph about the potential use of isotope composition. However, if one were to investigate the implications of residence time for the stable isotope composition, it would be more meaningful to perform the study using isotope-enabled general circulation models, so it remains unclear how this motivation fits to the present study design. The connection between the cited literature and the topic of the study remain vague and would require further discussion. For example, the residence time definition in Aggarwal et al., 2012, yields values ranging from 1 to 100 days, which are in contrast to the magnitude of residence time changes discussed here. The study of Markle et al, 2018 addresses in particular multi-centennial time scales, and it is not obvious without further discussion if their findings are applicable to the presented sensitivity experiments.

5. Title: See major issue 2 above - what is the significance of the statement that the WVL is increasing? Is the ratio IWV/P an accurate and pointed description for how the hydrolocial cycle will be experienced in the future, given that fluxes will intensify? Consider that you possibly could ease the struggle of motivating this study at present by de-emphasizing the WVL aspects. By essentially reversing the study setup, you could talk about IWV and P changes and their sensitivities first, and finally concluding by discussing to what extend the WVL can provide additional information or be mistaken as a confusing message (which is now implicitly stated in the introduction).

6. Now it is very difficult to compare the results from individual models in Fig. 1. Consider plotting at least panel a on a horizontal scale that emphasizes the differences, rather than bars for example as a set of box and whisker plots. Regarding the large number of individual models in panels a-b, it may be more useful to present results as histograms and move the individual model perspective into a supplement figure.

7. Conclusions: "If emissions evolve according to a business-as-usual pathway, the WVL could increase by 25% by the end of the 21st century because of the large expected temperature changes, and despite the projected aerosol emission reductions leading to a lower water vapour lifetime sensitivity."

   What are the implications of that conclusion? Does it actually matter if the WVL increases by 25%, what would be the consequences? The question if the residence time is a useful indicator to measure aspects of climate models remains ultimately unanswered. What do the differences between model mean states and their sensitivities signify? Is there actual value in using the residence time over inspecting total column water and precipitation/evaporation separately? Maybe that (open) question should be put into the focus of the introduction and answered in the conclusions. This is also related to major issue 2 and 5.

   Furthermore, given that models and observational estimates of the residence time disagree substantially, even on the same magnitude as the absolute changes predicted here (Trenberth, 2011; van der Ent and Tuinenborg, 2017), what is the overall uncertainty of these predicted changes? How do we know that the differences only can be interpreted meaningfully?

8. Pg. 8, L. 19: "a longer WVL implies a higher heavy isotope ratio ... and in turn indicates a larger fraction of convective vs. stratiform" - This conclusion seems to be based now on (weak) correlations between a set of isotope measurements in surface precipitation and climate variables. I suggest that a reliable extrapolation to future climate be based on the underlying physical processes instead.

**Minor comments**

Sec. 3.2 should in part be within methods section.

Sec. 3.4 needs a transition sentence from previous section/paragraph.

[Figure]

Pg. 2, L. 24: "although some argue for a substantially shorter lifetime of 4-5 days" - I believe the discussion in the literature argues revolves around a consideration of whether IWV/P accurately represents how long water resides in the atmosphere - rephrase.

Pg. 3, L. 21: Some details to this long list of referenced studies should be given.

At numerous locations throughout the manuscript, the writing is unclear. The reasons are varied, but often two or more ideas and arguments are complied into one sentence. Sometimes, within one sentence it is referred to several figures for different aspects. Below I list some of the places were writing can be substantially improved by careful editing:

Pg. 3, L. 9

Pg. 5, L. 13

Pg. 5, L. 22

Pg. 6, L. 2

Pg. 6, L. 31

Pg. 7, L. 1

Pg. 7, L. 15-29

Pg. 8, L. 18-19
* * *

---

## Author Comment (AC1) · 5 Jul 2019

**Response to reviewers**

**General response**

We are very grateful to all three reviewers for their valuable comments. All comments have been carefully considered and necessary changes made. Some of the comments from different reviewers were related and required larger changes to the manuscript, and we therefore list here the major and structural changes made. We feel that these changes have greatly improved the quality and importance of this manuscript.

- We have put more emphasis on changes in water vapour, and particularly changes in integrated water vapour between different climate drivers. This has been done by editing the title, abstract and conclusions, and by reversing the study setup so that changes in water vapour are discussed first, and changes in water vapour lifetime are discussed last. As a consequence, Figures 1-3 are now numbered 4-6 and vice versa.
- A new Section 3.5 has been added. This section includes a discussion of how changes in water vapour lifetime can be interpreted, and what the value of analysing changes in lifetime is. Some of the text that was previously in the introduction has now been included in this section.
- We have highlighted more the advantages of studying water vapour lifetime compared to analysing changes in IWV and P separately, and focused on implications of our results. This is now reflected in Section 3.5 and in the revised abstract and conclusions. A relevant reference, Singh et al. (2016), has been added to the manuscript.

**Reviewer #1**

*The authors present a study of water vapour lifetime, defined as the ratio of IVW/P, from a series climate models for present and future climate. The main driving forces of an increase of the lifetime in a warmer climate are analysed by a range of sensitivity studies. In addition to some major comments, the overall writing can be substantially improved in some sections. I therefore recommend major revisions.*

**Response:** We thank the reviewer for the very useful comments, which have helped improve the manuscript. Each comment has been carefully considered and necessary changes have been made. Please see general response above and point-by-point responses below.

**Major comments**

*1. Abstract: The abstract should be revised to better reflect the actual content of the manuscript.*

**Response:** The abstract has been substantially rewritten and should now well reflect the content and the reversed study setup.

*2. Introduction: "The fact that water vapour content increases more than precipitation with rising surface temperatures implies an expected increase in the lifetime (Douville et al., 2002; Held and Soden, 2006; Schneider et al., 2010), and hence a slowing down of the hydrological cycle. However, global-mean precipitation or evaporation fluxes are commonly referred to as the strength of the hydrological cycle, which, in contrast, implies an intensification or acceleration of the hydrological cycle with global warming. Douville et al. (2002) note that this conclusion is somewhat misleading because it suggests faster turnover of water, which is not the case. Hence, when the global hydrological cycle is said to intensify or accelerate with warming, it should be made clear that this refers to the fluxes and not the cycle as a whole" The takeaway from this paragraph is rather confusing. If the fluxes intensify, what is the significance of the overall slowdown? The authors need to resolve the more fundamental underlying issue of explaining the meaning or significance of a residence time change. If what matters for the impact of climate change is the intensification of fluxes, what is the purpose of talking about a slowdown of the hydrological cycle? Maybe there is a clear answer, but it needs to be stated somewhere early on to motivate the reader to adopt the perspective of IWV/P rather than IWV and P individually.*

**Response:** We agree and have moved most of this paragraph to the new Section 3.5 (please see general response). Here, the paragraph has been rewritten to make the takeaway message less confusing. We have also highlighted the importance of the slowdown and included a new and relevant reference (Singh et al., 2016). The end of this new paragraph now reads:

> "While the terminology could be confusing, both the amplification (through intensification of fluxes) and the slowdown (through longer lifetime) are important indicators of changes in the hydrological cycle. The intensification of fluxes means more precipitation globally and higher water availability, with potential consequences for extreme precipitation and water vapour feedback. The slowdown, however, is an important sign of changing precipitation patterns, since water vapour resides in the atmosphere for a longer time before precipitation, and this behaviour cannot be deduced based on analysing IWV and P separately. A longer water vapour lifetime implies an increased length scale of water vapour transport, so that the distance between evaporation and precipitation of moisture is greater, as has been shown in detail by Singh et al. (2016). They further mention implications of this increased transport length scale, such as the expansion of the Hadley circulation and a poleward shift of midlatitude precipitation maximum."

*3. Regarding the deep convective mass flux in the tropics: If the motivation of the study is to compare the convective mass flux between the different models, it may be relevant to consider additional quantities, such as convective vs. stratiform precipitation (noting that convective precipitation parameterisations have a large uncertainty and differ substantially between climate models), or the mass flux itself. As the study is designed presently, the mass flux and the implications thereof are rather implicit, and it remains unclear whether this quantity should be considered as an internal model variable or as the actual flux of mass as represented by climate models.*

**Response:** An analysis of the convective mass flux between the different models could provide some interesting results, but is clearly beyond the scope of this study, as the required diagnostics are not available from the simulations. The point on moisture transport is rather mentioned as an example of how WVL is an indicator of changes in dynamical processes in the hydrological cycle. After also considering the comments from the other reviewers (particularly comment no. 3 and 4 of reviewer #3), we have decided to remove the sentences describing the link between WVL and mass flux changes.

*4. On pg. 3, L. 10 onward, the authors state that "understanding the WVL has the potential to contribute to improved quantification of the hydrological cycle and its climate-induced changes", based on a previous paragraph about the potential use of isotope composition. However, if one were to investigate the implications of residence time for the stable isotope composition, it would be more meaningful to perform the study using isotope-enabled general circulation models, so it remains unclear how this motivation fits to the present study design. The connection between the cited literature and the topic of the study remain vague and would require further discussion. For example, the residence time definition in Aggarwal et al., 2012, yields values ranging from 1 to 100 days, which are in contrast to the magnitude of residence time changes discussed here. The study of Markle et al, 2018 addresses in particular multi-centennial time scales, and it is not obvious without further discussion if their findings are applicable to the presented sensitivity experiments.*

**Response:** The paragraph has now been moved to the new section 3.5 and made part of a discussion of the value of WVL. We agree that a study using isotope-enabled GCMs would be very interesting and could, in combination with observations, give information that may be used to improve parameterizations of vertical mass-exchange in the GCMs. However, isotope-enabled GCMs are so far limited, both in number of models and in the comprehensiveness of the isotope schemes (Aggarwal et al., 2016), and this makes such a study difficult to conduct. With the design of our study, we are instead able to focus on changes in water vapour and its lifetime for a range of GCMs and sensitivity experiments. Isotope-enabled GCMs are expected to become more common in the future, and we expect that this will increase the value of our work.

It is correct that the lifetimes presented in Aggarwal et al. (2012) (approx. 1-100 days) have a much wider range than in our study. The reason is that we focus on the global- and annual-mean lifetime (around 8 days) and lifetime changes while Aggarwal et al. present daily measurements from individual stations. We would also get a much wider range in the lifetimes from the models if we were to consider daily and local scales. The point is that the correlation between isotopes and water vapour lifetime is valid for all stations/seasons considered in Aggarwal et al., and these stations represent a range of climate regimes, from Addis Ababa in the tropics to Halley Bay in the Antarctic. Therefore, we can deduce that an increase in the global water vapour lifetime strongly indicates a higher heavy isotope ratio. The text specifies that the correlation between WVL and isotope ratio in precipitation is found for daily station measurements (and not necessary globally).

*5. Title: See major issue 2 above - what is the significance of the statement that the WVL is increasing? Is the ratio IWV/P an accurate and pointed description for how the hydrolocial cycle will be experienced in the future, given that fluxes will intensify? Consider that you possibly could ease the struggle of motivating this study at present by de-emphasizing the WVL aspects. By essentially reversing the study setup, you could talk about IWV and P changes and their sensitivities first, and finally concluding by discussing to what extend the WVL can provide additional information or be mistaken as a confusing message (which is now implicitly stated in the introduction).*

**Response:** This is a good idea. We agree and have now reversed the study setup, included a discussion of the potential importance of WVL, and changed the title (please see general response). The title is now "Water vapour adjustments and responses differ between climate drivers".

*6. Now it is very difficult to compare the results from individual models in Fig. 1. Consider plotting at least panel a on a horizontal scale that emphasizes the differences, rather than bars for example as a set of box and whisker plots. Regarding the large number of individual models in panels a-b, it may be more useful to present results as histograms and move the individual model perspective into a supplement figure.*

**Response:** We agree that it is difficult to compare the models in Fig. 1a and the rightmost plot in Fig. 1b (now Fig. 4a-b) but we would like to keep the individual models in the figure in the main manuscript. We have now narrowed the scale in two of the plots (WV lifetime and WV lifetime sensitivity), and this made it much easier to compare results from the different models.

*7. Conclusions: "If emissions evolve according to a business-as-usual pathway, the WVL could increase by 25% by the end of the 21st century because of the large expected temperature changes, and despite the projected aerosol emission reductions leading to a lower water vapour lifetime sensitivity." What are the implications of that conclusion? Does it actually matter if the WVL increases by 25%, what would be the consequences? The question if the residence time is a useful indicator to measure aspects of climate models remains ultimately unanswered. What do the differences between model mean states and their sensitivities signify? Is there actual value in using the residence time over inspecting total column water and precipitation/evaporation separately? Maybe that (open) question should be put into the focus of the introduction and answered in the conclusions. This is also related to major issue 2 and 5. Furthermore, given that models and observational estimates of the residence time disagree substantially, even on the same magnitude as the absolute changes predicted here (Trenberth, 2011; van der Ent and Tuinenborg, 2017), what is the overall uncertainty of these predicted changes? How do we know that the differences only can be interpreted meaningfully?*

**Response:** The conclusions have now been rewritten and a new Section 3.5 on the potential importance of WVL has been added (please see general response). Regarding the modelled vs. observed water vapour lifetimes, a discussion of uncertainties has been added to the new Section 3.5, also accounting for comment no. 4 of reviewer #3:

> "Among the most important caveats with our WVL findings is that climate models have known deficiencies, such as problems with representing vertical convective mass fluxes (Bony et al., 2015), surface moisture fluxes and entrainment/detrainment rates. Part of the reason is that GCMs have relatively coarse resolution and many processes, such as convection, need to be parameterized. However, we use a large multi-model ensemble with horizontal resolutions ranging from 1.4°×1.4° (MIROC-SPRINTARS) to 2.8°×2.8° (CanESM2), and model spread in future WVL change is lower (relative standard deviation (RSD) of 22%) than for, e.g., precipitation change (RSD of 30%) (Fig. 4). Nevertheless, compared with present-day WVL from reanalysis, the climate models have too short WVLs (Trenberth et al., 2011; see also Section 3.3). Kao et al. (2018) compared trends in precipitation and column water vapour data from 13 CMIP5 models with observational datasets and also found differences in the moisture recycling rate between observations and the CMIP5 models, and concluded that this discrepancy was caused by relatively poor simulations of precipitation. However, the long-term trend and inter-annual variability of column water vapour was very well captured by nearly all models."

*8. Pg. 8, L. 19: "a longer WVL implies a higher heavy isotope ratio ... and in turn indicates a larger fraction of convective vs. stratiform" - This conclusion seems to be based now on (weak) correlations between a set of isotope measurements in surface precipitation and climate variables. I suggest that a reliable extrapolation to future climate be based on the underlying physical processes instead.*

**Response:** The conclusions have been rewritten and the connection between isotopes and precipitation type has been removed.

**Minor comments**

*Sec. 3.2 should in part be within methods section.*

**Response:** We agree that part of this section could belong to the methods section. However, we think that it is easier for the reader to interpret the results when explained in relation to the figures, and therefore prefer to present Figures 5-6 at the same time as explaining the methodology used.

*Sec. 3.4 needs a transition sentence from previous section/paragraph.*

**Response:** This has now been added (Section 3.4 is now Section 3.2):

> "In order to explore reasons for differences in IWV between the drivers, we compare vertical profiles of specific and relative humidity and temperature for each of the climate drivers (Fig. 2)."

*Pg. 2, L. 24: "although some argue for a substantially shorter lifetime of 4-5 days" - I believe the discussion in the literature argues revolves around a consideration of whether IWV/P accurately represents how long water resides in the atmosphere - rephrase.*

**Response:** Good point. The sentence has now been changed to:

> "Studies identify a global-mean water vapour lifetime of 8-9 days for present-day conditions (van der Ent and Tuinenburg, 2017), although some argue that water only resides in the atmosphere for about 4-5 days (Laderach and Sodemann, 2016)."

*Pg. 3, L. 21: Some details to this long list of referenced studies should be given.*

**Response:** We have now reduced the long list of references and given some details to two of the studies. It now reads:

> "... but most of these studies have focused mainly on precipitation (e.g., Andrews et al., 2010; Bala et al., 2010; Xie et al., 2013; Samset et al., 2016). Recently, new insight into precipitation changes has been given by analysing the atmospheric energy budget (Richardson et al., 2018b) and by the use of radiative kernels (Myhre et al., 2018a)."

*At numerous locations throughout the manuscript, the writing is unclear. The reasons are varied, but often two or more ideas and arguments are complied into one sentence. Sometimes, within one sentence it is referred to several figures for different aspects. Below I list some of the places were writing can be substantially improved by careful editing:*

**Response:** We have now improved the writing in several of the places mentioned:

*Pg. 3, L. 9*        **Response:** Sentence removed because of other comments.

*Pg. 5, L. 13*        **Response:** Sentence split in two.

*Pg. 5, L. 22*        **Response:** Sentence split in two.

*Pg. 6, L. 2*        **Response:** Sentence slightly rewritten.

*Pg. 6, L. 31*        **Response:** Sentence rewritten.

*Pg. 7, L. 1*        **Response:** Sentence split in two.

*Pg. 7, L. 15-29*   **Response:** Many of the sentences rewritten and/or split in two.

*Pg. 8, L. 18-19*   **Response:** Sentence removed because of other comments.

**Reviewer #2**

*A catalog of idealized climate model experiments to assess precipitation response to different radiative forcings is here exploited to investigate how the atmospheric lifetime of water vapor is affected. This is diagnosed as integrated water vapor divided by precipitation rate which effectively characterizes how long it would take to precipitate out all the water vapor in the atmospheric column. Although it is obvious, based on past research, that this lifetime should increase, since thermodynamic and energetic constraints cause water vapor to increase at a faster fractional rate than precipitation, this work provides a useful investigation into the differences in this response between forcing agents, relating to fast adjustments and slow response to temperature, and further explores regional contributions. The most novel aspect, in my view, may be demonstrating how water vapor adjustments and responses differ between forcing agents. I recommend emphasizing this and I consider that this work merits publication following consideration of the suggestions below including the possibility of comparing with observed responses.*

**Response:** We thank the reviewer for the very useful comments, which have helped improve the manuscript. Each comment has been carefully considered and necessary changes have been made. Please see general response above and point-by-point responses below. In particular, we have now emphasized more the different water vapour adjustments. This has been done by reversing the study setup, as suggested by Reviewer #1, so that water vapour changes are discussed first and the water vapour lifetime changes are discussed last.

***Specific points***

*1) p.1, L24-26: the first 2 lines do not make much sense to me in the abstract and have marginal relevance to the results. Something outlining what water vapor lifetime is and why it is important would be more useful I think.*

**Response:** The abstract has been substantially rewritten, and the first two lines have been removed, to highlight more the different water vapour adjustments rather than the lifetime differences.

*2) p.1, L29: "projected" –> "simulated" (1986-2005 is not a projection)*

**Response:** Corrected.

*3) p.1, L31: "slows down the hydrological cycle" - if precipitation is increasing, the hydrological cycle could be thought of as speeding up since water is fluxing between atmosphere and surface more quickly so I suggest removing this confusing terminology.*

**Response:** Removed.

*4) p.1, L34 - "fast responses" should be clarified*

**Response:** This has now been defined by changing the beginning of the sentence to:

> "Fast responses, which include the initial radiative effect and rapid adjustments to an external forcing…"

*5) p.2, L18-20 - Is there a difference between water vapor residence time, lifetime and recycling rate (e.g. Li et al. 2011; Kao et al. 2018; van der Ent & Tuinenburg (2017); Allan & Zveryaev (2011) IJOC http://doi.org/10.1002/joc.2070). This could be clarified. Regional responses in water vapor lifetime may be misleading since the precipitation can result from transport of moisture from outside of the region and so not really reflect recycling rate within a box*

**Response:** This has now been clarified and the text now reads:

> "The water vapour lifetime is also known as the residence time and is commonly expressed as the ratio between the time-averaged global-mean integrated water vapour and precipitation (Trenberth, 1998; Douville et al., 2002; Bosilovich et al., 2005; Schneider et al., 2010; Kvalevåg et al., 2013). The water vapour recycling rate is the inverse of the lifetime (P/IWV) and most often expressed regionally rather than globally (Li et al., 2011; Kao et al., 2018), in which another factor is how much of the regional precipitation results from transport of water vapour from outside the region."

*6) p.4 L26 - RCP8.5 is a high emissions scenario but cannot simply be described as a business as usual pathway.*

**Response:** This has now been changed to "a high emission pathway".

*7) p.5, L15 - is an increase in WVL detectable in the historical period 1986-2005? Using trends from Allan et al. (2014) Surv. Geophys. http://doi.org/10.1007/s10712-012-9213-z for 1988-2008 and assuming WVL=8.9 days:*

*WVLS = WVL((1/IWV)(dIWV/dT) - (1/P)(dP/dT)) = 8.9x(0.064-0.028) = 0.32 days/K,*

*which is smaller than simulated perhaps due to additional noise from internal variability (with a large uncertainty). Alternatively:*

*dWVL/dt=WVL((1/IWV)(dIWV/dt) - (1/P)(dP/dT)) = 8.9x(0.0084 - 0.0018) = 0.06 days/decade (rather small)*

**Response:** We agree that it would be good to compare modelled WVL increases to observed values over the historical period, but the period 1986-2005 is short and likely heavily influenced by natural variability. What we show in the manuscript are modelled values between pre-industrial times (1850-1869) and recent times (1986-2005). Ideally, we should have had observed values between the same two time periods, but observations are too sparse to derive this.

*8) Fig.2 - additional annotation to show the meaning of dark/light bars in (a) and (b) would help the reader.*

**Response:** We agree and have now included legends both in plot (a) and (b) to show the meaning of the dark/light bars.

*9) Fig.3 - it is not clear from the scattering aerosol bar how the light and dark part contribute. Perhaps the total can be distinguished as a thick horizontal line or symbol (at the top of most bars but at -0.01 for scattering aerosol).*

**Response:** Good suggestion. We have now added a symbol to show the total, and included this in the legend.

*10) Fig. 4 - it would be more informative for me to group all the WV, E and P lines into 3 separate plots so that they can be compared across forcing agents. Are zonal values calculated using zonal dT or global dT?*

**Response:** Both ways of presenting the results have their advantages, but we agree that comparing the results across forcing agents are more informative and have changed the plot accordingly. In addition, we included a similar plot for the slow response, so that readers can compare differences between slow and total responses without referring to the supplementary. However, we prefer to keep the original Supplementary Figure S2 unchanged so that the results also can be compared across variables (it also makes less sense to compare across forcing agents when the results have not been normalized with the global dT). With the new plots, it does not make sense to include the global mean values next to each line (the plot would be too busy), so we have added a Table 1 with these values and edited the text accordingly. Zonal values are normalized using global dT – this has now been specified in the figure caption.

*11) p.7, L4-6 seems an important result and some more mechanistic discussion of this would be useful. Is the SO4 slow response small due to forcing predominantly affecting land which has less moisture availability? Or does this relate to the vertical temperature changes and the temperature dependence of the Clausius Clapeyron equation? Does the low level relative humidity increase explain the large fast response in BCx10 and why? On the other hand is this all explained by land-ocean temperature responses as implied? This could be summarized in the conclusions along with implications (why do we care?).*

**Response:** This is an interesting point and we agree that further discussion of these results would be useful. The small SO4x5 slow response and the large Sol+2% slow response is related to the vertical temperature changes and the temperature dependence of the Clausius Clapeyron equation. This can be seen in Fig. S7 and we have added/modified the text to explain this (a sentence has also been added in the conclusions to highlight the different slow responses between drivers):

> "Changes in specific humidity profiles for the slow response (Fig. 2b) show that the assumption of constant relative humidity does hold. When normalized with $\Delta T_s$, the specific humidity profiles are similar between the drivers, with a small exception for SO4x5, which shows a smaller increase throughout the troposphere. One reason is that SO4x5 is the driver that gives the least change in the temperature profile (and lapse rate up to 300 hPa) when normalized with surface temperature change (Fig. S7, lower left), and therefore the least change in the water vapour availability. It is also worth noting the strong difference between land and sea in the temperature change profile for this driver. The small increase in vertical temperature profiles compared to the other drivers could explain why SO4x5 has the smallest slow IWV response (Table 1). Sol+2%, which have the largest slow IWV response, has the second strongest increase in temperature profile per K surface temperature change (Fig. S7). BCx10 gives the strongest increase in atmospheric temperature, but a decrease in relative humidity, especially over land (Fig. S7), leads to a discrepancy between the actual specific humidity change and the temperature-driven change between the surface and 800 hPa (Fig. 2b)."

The low-level RH increase for BCx10 certainly contributes to the large fast response in IWV for this driver, as can be seen from Fig. 2a. However, it is difficult to know from the current simulations how much this contributes compared to the contribution due to warming of the atmospheric column. We have added/modified the text as follows:

> "In contrast to the CO2x2 experiment, BCx10 mostly yields a small increase in relative humidity (Fig. 2c), especially close to the surface, and therefore the specific humidity change for BCx10 is larger than the temperature-induced change throughout most of the troposphere (Fig. 2a). This low-level relative humidity increase contributes to the large fast IWV response for BCx10 (Table 1). Additional contributions come from atmospheric solar absorption due to BC, which leads to rapid atmospheric temperature increase (Fig. 2d) and therefore increased water vapour availability."

For CO2x2, the situation is opposite (RH decrease leading to small IWV fast response) and we have added a sentence to point this out:

> "This relative humidity decrease, and thus specific humidity decrease, contributes to the small fast IWV response for CO2x2 (Table 1)."

*12) p.7, L26 "small exception for SO4x5." Please be more explicit in what is meant.*

**Response:** The end of the sentence has now been expanded to explain this:

> "…, with a small exception for SO4x5, which shows a smaller increase throughout the troposphere."

*13) P.7, L30 - there is very little mention of Figure 6. Either this can be removed or a little more discussion of the Figure panels included.*

**Response:** The discussion of Fig. 6 (now Fig. 3) gives new insight into the fast vs. slow response of near-surface relative humidity to global warming. Therefore, we would like to keep the figure and have now included a bit more discussion and a sentence about the findings in the conclusions.

*14) Fig.5 - dashed=Clausius Clapeyron in the legend would help. It is difficult to see dashed and solid in (b) so perhaps this can be replaced with a relative humidity change plot.*

**Response:** The legend has been updated to show the meaning of the dashed lines. Regarding Fig. 5b (now Fig. 2b), we prefer to keep the specific humidity plot (a RH plot is shown in Fig. S7 in the Supplementary), but we have included an inset plot, which is zoomed in over the 700-1000 hPa region and shows the differences between dashed and solid lines much better.

*15) p.8 (Conclusions) - what is the significance of changes in water vapor lifetime above the differing fractional responses of P and IWV and implications for changes in the tropical circulation mass flux and precipitation intensity distribution, which is well known? Emphasizing what is novel will help increase the impact of this work.*

**Response:** The conclusions have now been rewritten with a stronger focus on the changes in integrated water vapour, and with more emphasis on the implications of changes in the water vapour lifetime (please also see general response).

**Reviewer #3**

*A number of climate model simulations from the CMIP5 intercomparison are used in order to estimate the change in water vapor lifetime with climate change. Water vapor lifetime is shown to increase by about 2 days in the next 100 years. Contributions from different climate drivers are analyzed using simulations from the Precipitation Driver Response Model Intercomparison Project (PDRMIP). Estimates for the combination of all drivers for the past are shown to be consistent with CMIP5 results. Changes in WVL are split into fast and slow responses. Changes in IWV per surface temperature change of different climate drivers are compared to the theoretical 7%/K increase that is expected assuming relative humidity to stay constant. BC shows the strongest increase in water vapor lifetime. The findings are very interesting but the paper is too concise to appreciate results fully. More information, explanations for assumptions and discussion needs to be added.*

**Response:** We thank the reviewer for the very useful comments, which have helped improved the manuscript. Each comment has been carefully considered and necessary changes have been made. Please see point-by-point responses below (and general response above).

*1. You calculate contributions from changes in IWV and P to ΔWVL by calculating the ΔWVL twice, with the IWV and P terms held constant one at a time (page 5 line 9-10). This means that you neglect nonlinear terms which needs to be mentioned. It is difficult to judge from the material presented if this is a good assumption, since figure 2a gives the fast WVLS and figure 2b the WVL itself. I suggest plotting the overall WVL change in figure 2b additionally.*

**Response:** This is a good point. The contribution to the fast $\Delta WVL$ from changes in each of IWV and P is calculated for each model as follows (using CO2x2 as an example):

$$\Delta WVL_{IWV} = \frac{IWV_{CO2x2}}{P_{base}} - \frac{IWV_{base}}{P_{base}} = WVL_{IWV,CO2x2} - WVL_{base}$$

$$\Delta WVL_{P} = \frac{IWV_{base}}{P_{CO2x2}} - \frac{IWV_{base}}{P_{base}} = WVL_{P,CO2x2} - WVL_{base}$$

In Fig. 2b (now Fig. 5b), the two terms have been scaled so that the sum of $\Delta WVL_{IWV,scaled}$+ $\Delta WVL_{P,scaled}$ equals the actual fast $\Delta WVL$:

$$\Delta WVL_{IWV,scaled} = \Delta WVL_{IWV} \frac{\Delta WVL}{\Delta WVL_{IWV} + \Delta WVL_{P}}$$

$$\Delta WVL_{P,scaled} = \Delta WVL_{P} \frac{\Delta WVL}{\Delta WVL_{IWV} + \Delta WVL_{P}}$$

To test how good our assumption is, we need to check whether $\Delta WVL_{IWV}$+ $\Delta WVL_{P}$ is notably different from $\Delta WVL$. The following figure shows $\Delta WVL_{IWV,scaled}$+ $\Delta WVL_{P,scaled}$ (left plot) vs. $\Delta WVL_{IWV}$+ $\Delta WVL_{P}$ (right plot) for each model and driver:

[Figure]

[Figure]

The plots show that differences are very small. In the model mean, the largest difference is for BCx10: (0.4288-0.4222 days)/0.4222 days = 1.56%. Based on this, the following sentence has been added at the end of Section 2.3 in the manuscript:

> "This assumption involves nonlinear terms, but the model-mean difference between the actual $\Delta WVL$ and the sum $\Delta WVL_{IWV}$+$\Delta WVL_{P}$ is less than 2% for all drivers."

*2. Could you please give an explanation why it makes sense to scale ΔWVL with RF (page 6 line 17).*

**Response:** We have added a sentence to clarify this:

> "By scaling the fast ΔWVL with RF, we are able to estimate the historical contribution to ΔWVL from each driver, in a similar way to what has been done before for other quantities (e.g., sensible heat flux changes in Myhre et al. (2018b))."

*3. Water vapor lifetime is increased which is supposed to be connected with a decrease in vertical mass fluxes. But a decrease in vertical mass fluxes should be connected with a moistening of the lower troposphere which appears not to be the case. Is there an explanation for this behavior?*

**Response:** An analysis of the convective mass flux in relation to the water vapour lifetime could provide some interesting results, but is clearly beyond the scope of this study, as the required diagnostics are not available from the simulations. The point on moisture transport is rather mentioned as an example of how WVL is an indicator of changes in dynamical processes in the hydrological cycle. After also considering the comments from the other reviewers (particularly major comment no. 3 of reviewer #1), we have decided to remove the sentences describing the link between WVL and mass flux changes.

*4. Changes of water vapor lifetimes are connected with vertical mass fluxes. For the analysis of WVL changes you use climate models which have problems representing those mass fluxes. In particular convective mass fluxes are known to be a source of large uncertainty within climate models. Surface moisture fluxes may also be problematic. Vertical profiles of humidity may be strongly dependent on entrainment and detrainment rates which are highly problematic. It would be good to add a discussion about how dependent results are on known deficiencies in global models. Original model resolutions need to be given.*

**Response:** A discussion of uncertainties has been added to the new Section 3.5, also accounting for major comment no. 7 of reviewer #1:

> "Among the most important caveats with our WVL findings is that climate models have known deficiencies, such as problems with representing vertical convective mass fluxes (Bony et al., 2015), surface moisture fluxes and entrainment/detrainment rates. Part of the reason is that GCMs have relatively coarse resolution and many processes, such as convection, need to be parameterized. However, we use a large multi-model ensemble with horizontal resolutions ranging from 1.4°×1.4° (MIROC-SPRINTARS) to 2.8°×2.8° (CanESM2), and model spread in future WVL change is lower (relative standard deviation (RSD) of 22%) than for, e.g., precipitation change (RSD of 30%) (Fig. 4). Nevertheless, compared with present-day WVL from reanalysis, the climate models have too short WVLs (Trenberth et al., 2011; see also Section 3.3). Kao et al. (2018) compared trends in precipitation and column water vapour data from 13 CMIP5 models with observational datasets and also found differences in the moisture recycling rate between observations and the CMIP5 models, and concluded that this discrepancy was caused by relatively poor simulations of precipitation. However, the long-term trend and inter-annual variability of column water vapour was very well captured by nearly all models."

---

## Author Response (AR2)

**Response to reviewers**

**Reviewer #1**

*Panels in Fig. 3 are very small*

**Response:** The panels in Figure 3 have now been enlarged.

**Reviewer #3**

*I have only two further comments:*

*1. uncertainty due to low resolution models: What do you mean with 'Part of the reason is that GCMs have relatively coarse resolution and many processes, such as convection, need to be parameterized. However, we use a large multi-model ensemble with horizontal resolutions ranging from 1.4°×1.4° (MIROC-SPRINTARS) to 2.8°×2.8° (CanESM2),'. With this kind of resolution convection is not resolved which means that, if a decrease in convective mass fluxes is a major reason for the increase in WVL, then the results are connected with a large uncertainty. I think it is only fair to the reader to spell this out.*

**Response:** We agree that this sentence was not well formulated and have changed the text accordingly:

> "Part of the reason is that GCMs have relatively coarse resolution and many processes, such as convection, need to be parameterized. The model spread in future WVL change is lower (relative standard deviation (RSD) of 22%) than for, e.g., precipitation change (RSD of 30%) (Fig. 4), but the horizontal resolutions in the PDRMIP models range from 1.4°×1.4° (MIROC-SPRINTARS) to 2.8°×2.8° (CanESM2), where convection needs to be parameterized. This means that the uncertainty is larger if a decrease in convective mass fluxes is a major reason for the increase in WVL."

*2. The forcing component BC has a large impact on the results. But it appears to me that nowhere in the paper it is mentioned by which physical processes BC acts on WV and P. I assume that we are not only talking about direct radiative effects but indirect aerosol cloud effects as well. This and the fact that indirect aerosol cloud effects are connected with a large uncertainty should be mentioned somewhere.*

**Response:** We agree and have added this information in the Methods section:

[revised manuscript text omitted]